# Investigation on the Edge Chipping in Ultrasonic Assisted Sawing of Monocrystalline Silicon

**DOI:** 10.3390/mi10090616

**Published:** 2019-09-16

**Authors:** Jianyun Shen, Xu Zhu, Jianbin Chen, Ping Tao, Xian Wu

**Affiliations:** College of Mechanical Engineering and Automation, Huaqiao University, Xiamen 361021, China; jianyun@hqu.edu.cn (J.S.); hqucjb@163.com (J.C.); 17013080031@hqu.edu.cn (P.T.); xianwu@hqu.edu.cn (X.W.)

**Keywords:** monocrystalline silicon, ultrasonic assisted sawing, edge chipping, edge chipping mechanism

## Abstract

Monocrystalline silicon is an important semiconductor material and occupies a large part of the market demand. However, as a hard-brittle material, monocrystalline silicon is extremely prone to happen edge chipping during sawing processing. The edge chipping seriously affects the quality and performance of silicon wafers. In this paper, both conventional and ultrasonicassisted sawing tests were carried out on monocrystalline silicon to study the formation mechanism of edge chipping. The shape and size of edge chipping after sawing process were observed and measured. The experimental results demonstrated that different sawing processes present different material removal modes and edge quality. The mode of crack propagation was continuous cracks in conventional sawing process, while the expansion mode in ultrasonic assisted sawing was blasting microcracks. This results in the cutting force of ultrasonic assisted sawing becomes much smaller than that of conventional sawing process, which can reduce the size of edge chipping and improve the quality of machined surface.

## 1. Introduction

As an important semiconductor material, monocrystalline silicon is widely employed in electronics, communications and energy field [1,2,3,4]. With the development of semiconductor material technology, there are increasing demands on the performance and quality of silicon wafers, and the proportion of large-diameter silicon wafers that is suitable for micro fabrication in the market is increasing as well. At present, the production process of monocrystalline silicon components includes pulling, wire cutting, grinding, polishing, and dicing [5,6,7,8]. The dicing is the last step before packaging [9,10]. As a typical hard-brittle material, monocrystalline silicon is easy to produce brittle cracks and damages during the cutting process [11,12,13]. The occurrence of edge chipping will greatly reduce the performance and quality of semiconductor products.

The sawing process of silicon wafer with diamond saw blade is a common cutting method. Efra et al. [14] pointed out that the phenomenon of edge chipping in the cutting process is one of the three main factors that reduce production efficiency. Lin et al. [15] discovered that the surface features of saw blade were an important factor Affecting the phenomenon of edge chipping. Luo et al. [16] explored the edge chipping mechanism through studying the relationship between sawing direction, crystal direction and chip size during cutting of monocrystalline silicon with diamond saw blade. Moreover, Cvetkovic et al. [17] found that the ultra-precision cutting with the saw blade has many unique advantages for suppressing edge chipping and reducing sidewall roughness compared to wire sawing, but it is easy to cause greater sidewall curvature.

As an advanced machining technology, the ultrasonic vibration assisted sawing is widely used to machine hard-brittle materials [18,19,20,21]. It not only improves the surface quality after machining, but also increases the machining efficiency [22,23,24,25,26]. In addition, the sawing force during processing can be reduced, which is helpful to reduce edge chipping [27,28,29,30,31]. Wang et al. [32] designed a novel experimental method and studied the characteristics of edge chipping in rotary ultrasonic drilling. Shen et al. [33] developed a radial ultrasonic vibrating shank that could be mounted directly on the CNC center, and explored the groove morphology after sawing and the characteristics of the sawing force. Nowadays, ultrasonic vibration assisted sawing has become an important research direction in the field of micro fabrication of monocrystalline silicon wafers.

In this paper, ultrasonic assisted sawing technique was taken advantage to reduce the edge chipping of monocrystalline silicon. The formation mechanism of edge chipping during sawing process was analyzed in detail. Effects of ultrasonic assisted vibration on the edge chipping phenomenon during sawing were studied by monitoring the sawing force, measuring the edge chipping size and observing the edge morphology.

## 2. Kinematic Characteristics of an Abrasive Particle in Ultrasonic Assisted Sawing

Ultrasonic assisted sawing can be seen as a combination of conventional sawing and ultrasonic vibration processing. As shown in Figure 1, in the ultrasonic-assisted sawing, the motion of abrasive grains on the saw blade includes a rotational motion, an ultrasonic vibration perpendicular to the workpiece, and a motion in the feed direction. According to our previous study, the trajectory of single abrasive grain in ultrasonic assisted sawing can be expressed as [34]:(1)$$l_{u}\left(t\right)=\left(\begin{array}{c} l_{ux}\left(t\right)\\ l_{uy}\left(t\right) \end{array}\right)=\left(\begin{array}{c} r\sin\left(\frac{2\pi \omega t}{60}\right)+v_{\omega }t+A\sin(2\pi ft)\sin\left(\frac{2\pi \omega t}{60}\right)\\ r\left(1-\cos\left(\frac{2\pi \omega t}{60}\right)-A\sin(2\pi ft)\cos\left(\frac{2\pi \omega t}{60}\right)\right) \end{array}\right)$$
where *A* is the ultrasonic amplitude, *f* is the ultrasonic vibration frequency, $v_{w}$ is the feed speed of workpiece, *ω* and *r* are the angular velocity and radius of the saw blade, respectively, and *t* is the machining time. 

The trajectory schematic diagram when single abrasive grain rotating drawn using MATLAB software (R2014b, MathWorks, Inc., Natick, MA, USA) according to Formula (1) is shown in Figure 2a. As illustrated in Figure 2a, the cutting distance of grains in the ultrasonic assisted sawing is longer compared with conventional sawing in the same time. It is meant that abrasive grains in ultrasonic assisted sawing have greater speed compared to conventional sawing. Furthermore, the interaction relationship between abrasive grains and the workpiece is shown in Figure 2b. It can be found that the abrasive grains intermittently contact with the workpiece surface during cutting. That means the abrasive grains repeatedly cut-in and hammer the workpiece surface in the cutting zone of cutting arc with ultrasonic frequency. Effective contact time *t_e_* can be calculated according to the study by Wang et al. [35].
(2)$$t_{e}=\frac{1}{\pi f}\sqrt{\frac{2d_{p}}{A}}$$
where, *d_p_* is the penetration depth at which the abrasive grains are cut into the workpiece.

## 3. Mechanism of Edge Chipping in Ultrasonic Assisted Sawing

The phenomenon of edge chipping after sawing is mainly appeared on both sides of the machined groove, as shown in Figure 3. This brittle fracture damages can lead to a decrease in the quality of workpiece edge after cutting process and affect the dimensional accuracy. In order to analyze the formation mechanism of edge chipping during sawing, some assumptions and simplifications were based in this study. Firstly, the monocrystalline silicon in the test was regarded as an ideal brittle material, and there were no cracks and defects before processing. Then the shape of diamond abrasive grains on the cutting blade was regarded as a conical shape of the same shape and size. 

As mentioned in the analysis of the kinematics of abrasive grains, diamond particles that were held on the cutting blade repeatedly hammer the workpiece surface with ultrasonic frequency during the sawing process. Therefore, this composite motion characteristic of diamond particle has great effect on the final material removal mode. There are two material removal modes in ultrasonic assisted sawing process, which are brittle removal and ductile domain removal mode. And the brittle removal mode is the main factor that forms brittle fracture cracks on the workpiece surface.

The generation of edge chipping can be divided into three steps: crack generation, crack propagation and edge chipping. When the abrasive grains slightly contact with the workpiece, the abrasive grains are only scrapping and rubbing on the workpiece surface, which in turn produces a plastic deformation zone. With the diamond saw blade rotating, the sawing force increases as the cutting thickness of the abrasive grains increases. And then, the material removal mode changes to brittle fracture. Simultaneously, microcracks including median cracks and transverse cracks are generated. The median crack extends to the inside of the workpiece and forms internal damage of the machined surface. The extension of transverse crack will cause the material to be removed and ultimately determine the surface roughness of the workpiece after processing. As the lateral crack gradually expands to the workpiece surface, an edge-chipping phenomenon occurs. The median crack and transverse crack have a greatly effect on the material removal mode during ultrasonic assisted sawing. The extended width of the transverse crack directly affects the size of the edge chipping area. Therefore, the reduction of the extended length of the transverse crack can effectively reduce the size of the edge chipping.

Previous analyses have indicated that the suppression of transverse crack propagating is the key to reduce the size of the edge chipping. Meanwhile, the indentation behaviour of effective abrasive acting on the workpiece surface is the director factor for generating the transverse crack. Then, an idealized model was developed to illustrate the abrasive grain cutting into the workpiece surface and producing transverse cracks, as shown in Figure 4, where *θ* is the maximum depth when the single abrasive grain cutting into the work-piece surface (i.e., the depth of cutting *a_p_*), and *Ψ* is the tip angle of the diamond particle. The diamond particle is assumed to present a tip angle of 60 degrees and *T_c_* is the size of the transverse crack. *F_m_* is the impact force, reflected by normal force *F_n_* and tangential force *F_t_*, that is acting on the workpiece from the abrasive grain. Therefore, based on the research of Lambropoulos et al. [36] the *T_c_* of the transverse crack can be calculated as: (3)$$T_{c}=0.12{\left(\frac{E}{H_{v}}\right)}^{\frac{1}{3}}{\left(\cos\frac{\Psi }{2}\right)}^{\frac{4}{9}}{\left(\frac{F_{m}}{K_{IC}}\right)}^{\frac{2}{3}}$$
where *E*, *H_v_* and *K_IC_* are respectively elastic modulus, microhardness and fracture toughness. 

According to the basic principles of fracture mechanics and Equation (3), the size of edge chipping can be assumed to be positive related with the transverse crack propagating under the impact force of abrasive grain. Thus, the reduction of impact force is a critical factor to decrease the length of transverse crack.

## 4. Test Procedure

The sawing test was carried out on a HAAS OM-A2 machining center (Haas Automation Inc., Oxnard, CA, USA), which had a maximum spindle rotation of 30,000 rpm. The machine can provide a positional accuracy of ±1 μm, and its radial runout is less than 1 μm. As shown in Figure 5. The test device mainly includes an ultrasonic vibration system and a signal acquisition system. The ultrasonic system comprises an ultrasonic generator, a transducer and an amplitude transformer. In the test, the ultrasonic generator generates an ultrasonic electric signal, and the transducer converts it into ultrasonic mechanical vibration and amplified by the amplitude transformer to obtain the amplitude required for the test. The workpiece has been pre-polished to ensure the smooth initial surface. And its material property parameters are shown in Table 1. In the test, the workpiece was directly bonded to the end of the amplitude transformer. The tool used in the test was a diamond saw blade that was clamped to the tool shank and perpendicular to the work-piece surface. The saw blade has a diameter of 50 mm, a thickness of 0.3 mm, and a diamond particle size of 40 μm. 

A fixture of L shape was designed to hold the workpiece and ultrasonic vibration system on the dynamometer. Then, they were clamped together on the machine worktable as shown in Figure 5. The Kistler dynamometer 9257B was used to record the impact force that was reflected by normal force and tangential force during the sawing process. The measured impact force is a time-averaged value of the actual impacting force acting on the workpiece surface. The specific test parameters are shown in Table 2. Based on previous research, four different levels were selected for the line velocity, feed rate and cutting depth. The ultrasonic vibration parameters were fixed. Each experiment was repeated quantic to safeguard the accuracy of the measurements that could lead to measurement uncertainties. After the test, a scanning electron microscope was used to observe and measure the morphology and size of edge chipping.

## 5. Results and Discussion

### 5.1. The Impact Force

It has been pointed out in the analysis of the edge chipping mechanism that the magnitude of the impact force *F_m_*, including normal force *F_n_* and tangential force *F_t_*, has a direct influence on the edge chipping size. As illustrated in Figure 6, the impact force, such as normal force, was obtained from the acquired signals by calculating the difference value between the no-load and sawing process. Each experiment sawed five grooves in the same parameters and calculated their arithmetic mean to reduce random error between the measured sawing force and true force. 

Figure 7 shows the variation trend of impact force under different processing parameters. From Figure 7a and b it can be found that the variation trend of two sawing methods has the same law, the impact force increases with the increase of the sawing depth and the feed speed. Figure 7c shows the influence of the line speed on the sawing force for two feed rate, 240 mm/min and 320 mm/min, and a sawing depth of 60 µm. Instead, the impact force decreases with increasing line speed. This is owing to the increase of line speed increases the number of grains cutting the workpiece in unit time. The lower force could be obtained from ultrasonic assisted sawing in the same line speed due to its more faster speed, in unit time, as discussed in Section 2. This variation law means that the smaller the material removal amount of single abrasive grain, the smaller impact force, and the decrease of the impact force can reduce the extended length of the crack, thereby obtaining a smaller edge chipping size. 

By the way, it can be seen from the test results that, under the same processing parameters, the impact force under ultrasonic sawing is lower than that of conventional sawing. Because in ultrasonic assisted sawing, the linear velocity of the abrasives under the combined action of ultrasonic vibration and rotational motion is faster than that of conventional sawing, so that it produces a higher collision momentum at the moment when the abrasives cut the workpiece. This causes the monocrystalline silicon that is a brittle material to be more susceptible to microcracking and fracture, which is then removed by subsequent abrasive grains on a basis of fracture. The effect of ultrasonic vibration on the one hand transforms the continuity of cracks in conventional sawing into microcracking, and on the other hand changes the way of material removal and reduces the strength of the material. This change allows for smaller cutting forces and edge chipping sizes when ultrasonically assisted sawing of monocrystalline silicon, improving the surface quality of the edge of the monocrystalline silicon after processing.

The specific energy is an index to evaluate the energy that removed the material per unit volume during processing. It can reflect the interaction mechanism between abrasive particles and materials in the process of sawing. Obviously, as shown in the Figure 8, the specific energy decreases with the increase of material removal rate. But the specific energy decreases sharply with the increase of material removal rate in the conventional sawing. This is because in the conventional sawing, the abrasive particles are mainly scratched, rubbed, and squeezed on the surface of workpiece using the rake face. So that it presents elastic deformation at first, and then brittle fracture occurs when the material reaches its rupture strength limit and the material is eventually removed. With the increase of material removal rate, a large number of cracks are generated on the processed surface, which lead to the collapse of a large area. However, the required energy of brittle fracture is much lower than the energy consumed in the plastic removal process, so the specific energy is greatly reduced. But in the ultrasonic assisted sawing, the abrasive particles impact on the surface of workpiece under the vibration of ultrasonic high frequency, which dramatically increases the acceleration of abrasive particles. It promotes the monocrystalline silicon to be more prone to produce fatigue failure, which is convenient to remove the material. Thus, the energy consumption of material remove is decreased. 

### 5.2. The Edge Chipping Morphology

Figure 9 shows the groove topography that obtained by conventional sawing and ultrasonic sawing under the same processing parameters, *v_s_* = 10.5 m/s, *f_w_* = 80 mm/min, *a_p_* = 40 μm. It can be observed that the edge chipping area of the ultrasonic sawing is significantly smaller than that of the conventional sawing. In the conventional sawing process, the cutting process of the abrasive grains is continuous. When the brittle crack is generated, the crack will continuously expand under the action of the cutting force until it spreads to the surface of the workpiece. It commonly results in the final large-area edge chipping, as shown in Figure 9a. However, in ultrasonic sawing, the abrasive grains repeatedly cut in and cut out workpiece with ultrasonic frequency in the single contact arc. When the abrasive grains cut in, the local stress applied on the workpiece is very large due to the compound of cutting and ultrasonic vibration movement. With the large enough stress, the workpiece in the contact zone happens rapidly collapsing at very short time and generates many microcracks. When the abrasive grains cut out, the stress is decreasing, the collapsing phenomenon is stopped and then microcracks propagate just in a small distance. Subsequently, this process will go on periodically with the next ultrasonic vibration movement. This indicates that the material removal mode in ultrasonic sawing mainly is micro-crack bursting rather than continuous crack propagation in conventional sawing. The different material removal mode is helpful to prevent the continuous propagation of brittle cracks and results in a smaller edge chipping size, as shown in Figure 9b.

Figure 10 shows the edge chipping topography at high magnification. The larger edge chipping zone is clear to see in conventional sawing due to the continuous crack propagation, as shown in Figure 10a. This causes both the depth and width of edge chipping are very large. On the contrary, in the ultrasonic sawing, since the brittle crack stops propagation at the initial stage due to the local micro-crushing of material, the continuous crack propagation is suppressed with the cut in and cut out process. The obtained edge chipping zone is relatively smaller, as shown in Figure 10b. 

### 5.3. Theedge Chipping Size

Damage layer and scratches are still caused in process, whether ultrasonic assisted sawing or conventional sawing, following process must be adapted to eliminate them in order to obtain super-smooth and non-damage surface. Hence, the maximum size of edge chipping will directly determine the amount of material removal in subsequent processes. As presented in Figure 11, the maximum width from the edge of machined groove to the farthest point of edge chipping is used as a criterion to evaluate the edge chipping.

The influences of machining parameters on the edge chipping size are shown in Figure 12. It can be found that the varying trend of the edge chipping size of two sawing methods is consistent, but the edge chipping size of ultrasonic sawing is smaller than that of the conventional sawing under the same parameters. As shown in Figure 12a,b, the edge chipping size increases as the sawing depth and feed rate increase, but decreases as the line speed increases demonstrated in Figure 12c. From this it can be concluded that the smaller sawing depth, feed rate and line speed contribute to a smaller edge chipping size.

As mentioned earlier, edge chipping during sawing process is a brittle failure resulting from propagation of a crack, especially transverse crack, under the sawing force. Crack propagation characteristics and the sawing force during manufacturing are two major factors that determine the maximum size of edge chipping. The ultrasonic vibration assisted method prominently reduces the edge chipping size, especially in which case that the chipping size is large. As shown in Figure 12c, the smallest edge chipping size is obtained in the ultrasonic sawing with the processing parameters of *a_p_* = 60 μm, *f_w_* = 80 mm/min and *v_s_* = 34 m/s. As shown in Figure 12b, the largest edge chipping size is obtained in conventional sawing with the processing parameters of *a_p_* = 60 μm and *f_w_* = 320 mm/min, and *v_s_* = 10.5 m/s. In conventional sawing, the brittle crack propagates in a continuous manner, but the ultrasonic vibration causes the material collapsing and the brittle crack propagates just in a small distance with the cut in and cut out process. This effect suppresses the large-area continuous expansion of the brittle crack and greatly reduces the edge chipping size.

## 6. Summary and Conclusions

In this paper, the ultrasonic assisted sawing of monocrystalline silicon were performed to reduce the edge chipping size. The mechanism of reducing edge chipping size during ultrasonic assisted sawing was derived from detailed observation of the morphology of edge chipping and the sawing force. The following conclusions are drawn from the experimental results: During the sawing process, when the transverse crack propagates to the workpiece surface, the edge chipping phenomenon occurs due to brittle fracture of the material. The impact force of abrasive particle on the workpiece has great effect on the transverse crack propagating and is the key factor that determinate the edge chipping size. Hence, reducing impact force is helpful to reduce the edge chipping size.The vibration in the ultrasonic assisted sawing changes the material removal mechanism to microcrack blasting mode compared to the continuous crack propagation in conventional sawing. This change has led to the brittle crack propagation stopped at the initial stage and just expended in a small distance, which reduces the maximus size of edge chipping. Simultaneously, the reduction of sawing force further decreases the length of crack propagation, especially transverse crack, during the sawing process. The combination of two factors would get the smaller size of edge chipping.

## Figures and Tables

**Figure 1 micromachines-10-00616-f001:**
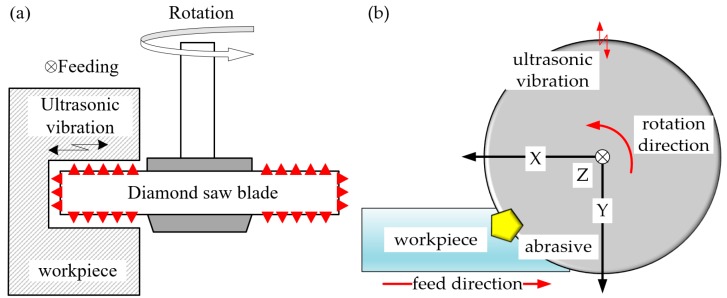
Illustration of ultrasonic-assisted sawing. (**a**) basic configuration; (**b**) the motion of single abrasive grain.

**Figure 2 micromachines-10-00616-f002:**
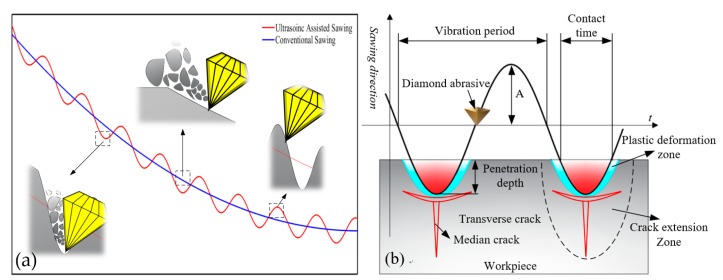
Simulation of an abrasive trajectory: (**a**) Movement track of abrasive grain during ultrasonic assisted sawing; (**b**) Cutting process of a single abrasive in the cutting arc area of ultrasonic sawing cutting.

**Figure 3 micromachines-10-00616-f003:**
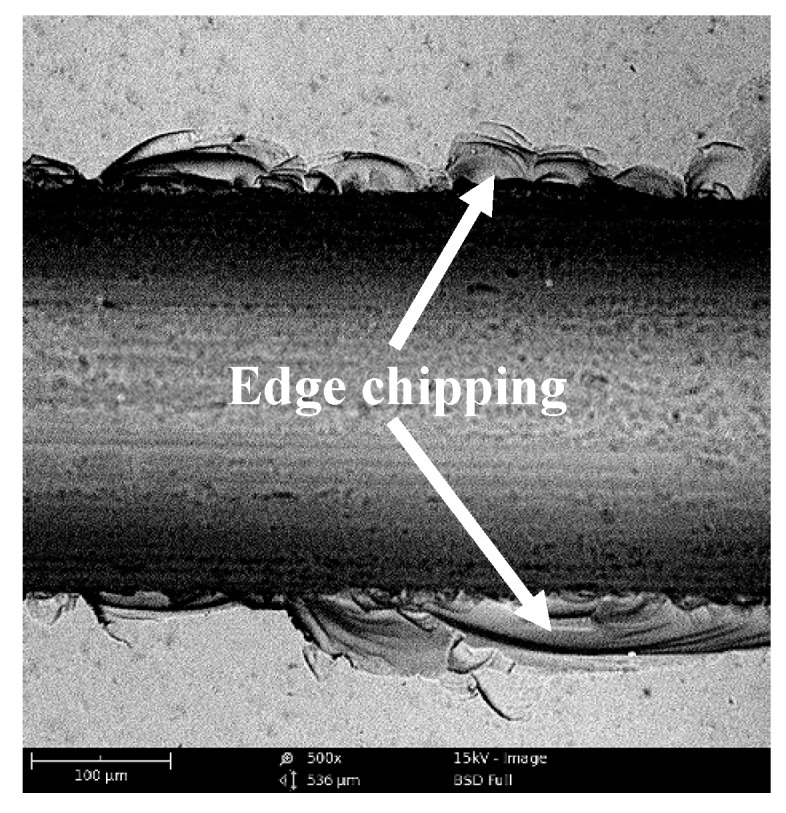
The edge chipping in sawing.

**Figure 4 micromachines-10-00616-f004:**
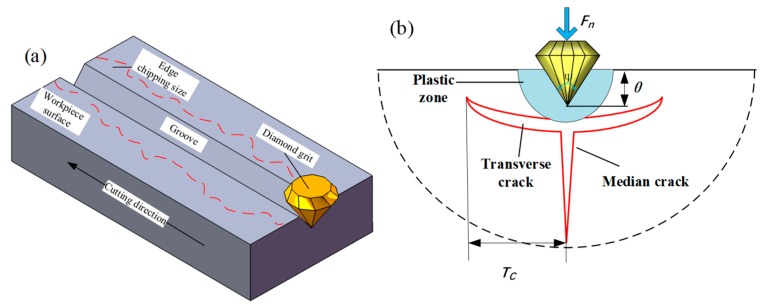
The formation of edge chipping. (**a**) a machined groove by single abrasive grain; (**b**) damaged induced by sawing.

**Figure 5 micromachines-10-00616-f005:**
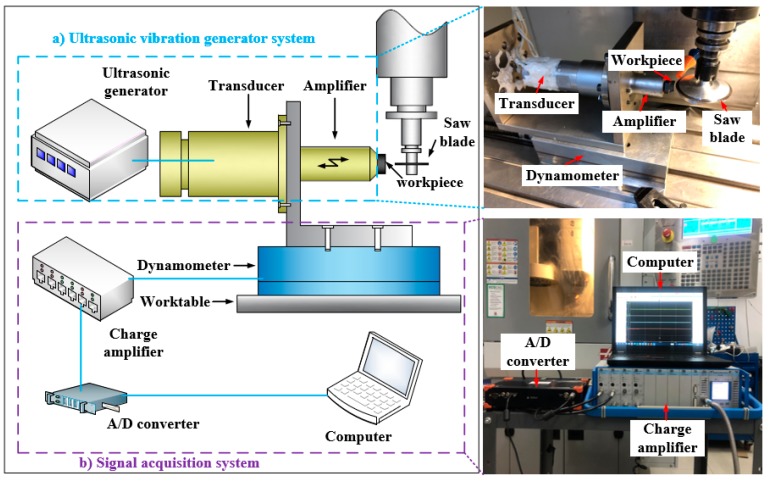
The schematic of ultrasonic assisted sawing experiment. (**a**) Ultrasonic vibration generator system; (**b**) Signal acquisition system.

**Figure 6 micromachines-10-00616-f006:**
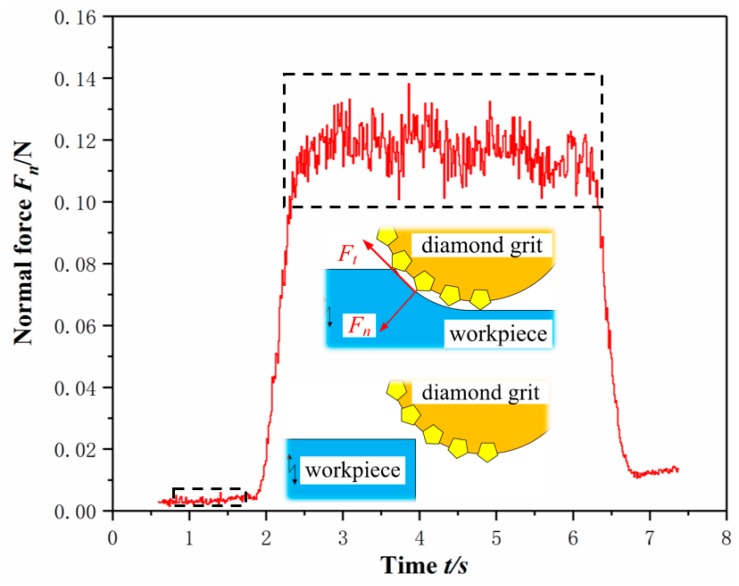
Typical curve of sawing force in feeding direction versus sawing time.

**Figure 7 micromachines-10-00616-f007:**
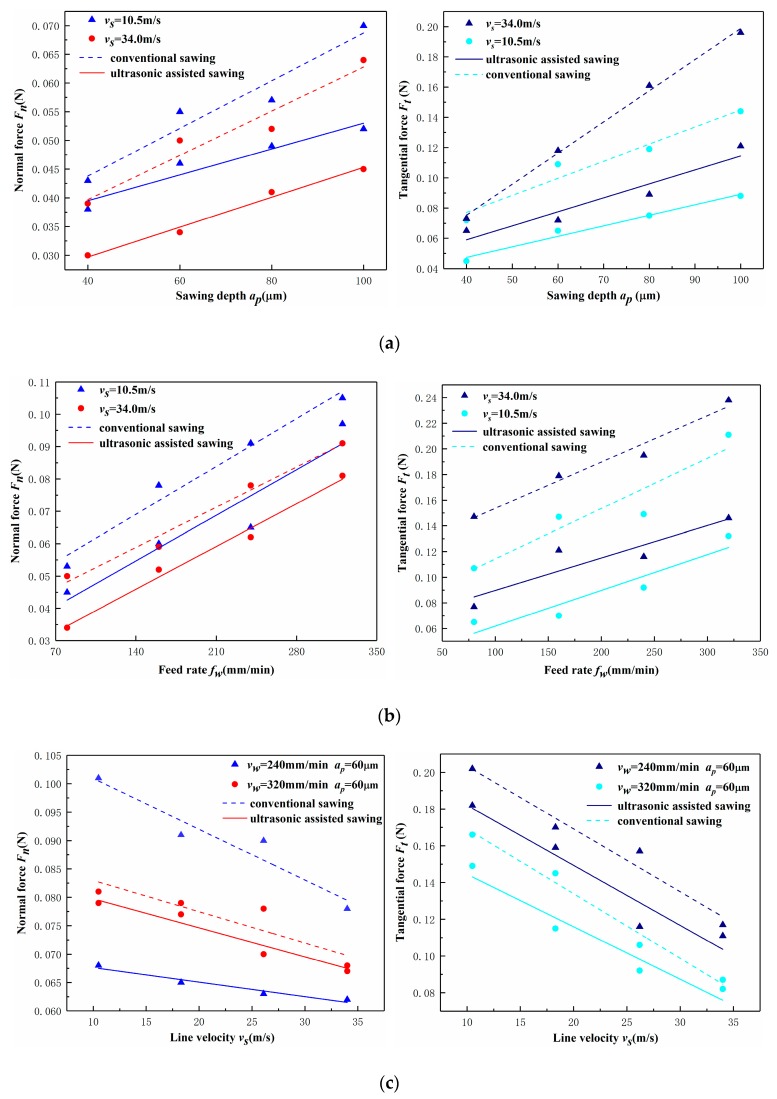
The impact force *F_m_* varying with machining parameters: (**a**) The impact force versus sawing depth (*v_w_* = 80 mm/min); (**b**) The impact force versus feed rate (*a_p_* = 60 μm); (**c**) The impact force versus line velocity.

**Figure 8 micromachines-10-00616-f008:**
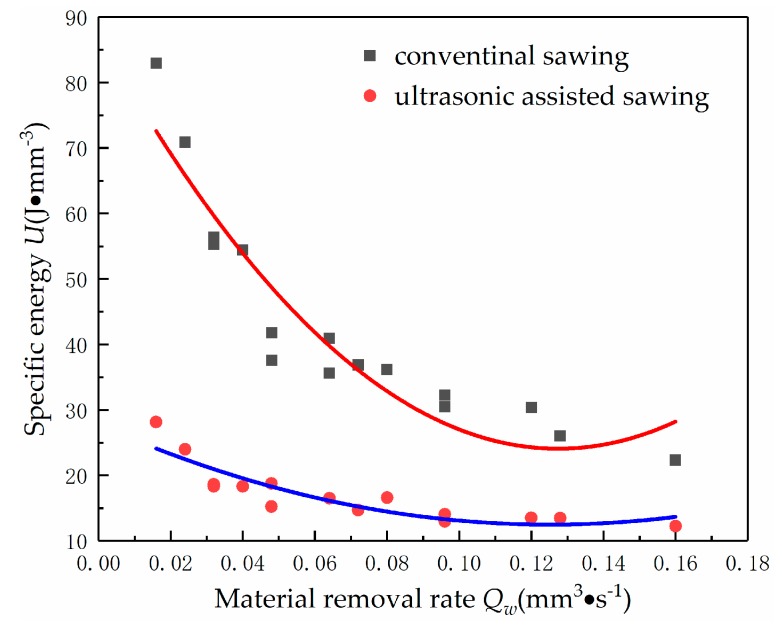
The specific energy *U* varying with material removal rate *Q_w_*.

**Figure 9 micromachines-10-00616-f009:**
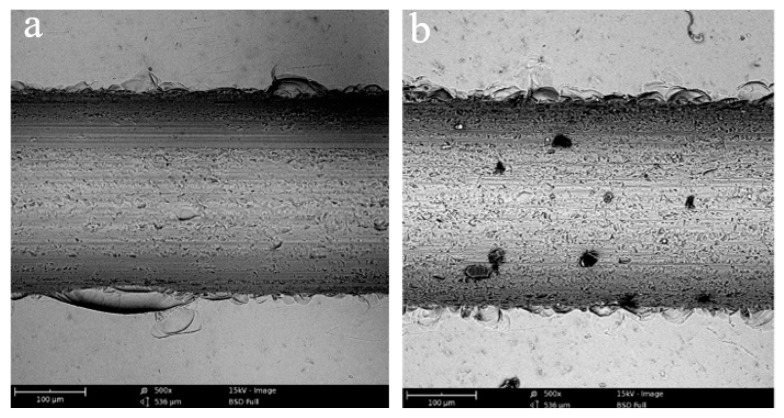
The morphology of edge chipping: (**a**) conventional sawing; (**b**) ultrasonic sawing.

**Figure 10 micromachines-10-00616-f010:**
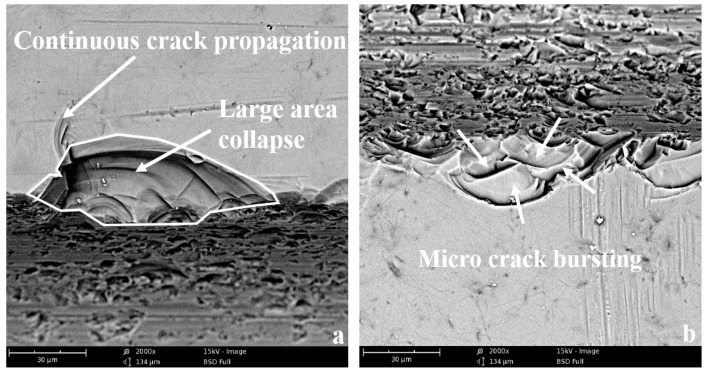
The comparison of material removal mode: (**a**) conventional sawing; (**b**) ultrasonic sawing.

**Figure 11 micromachines-10-00616-f011:**
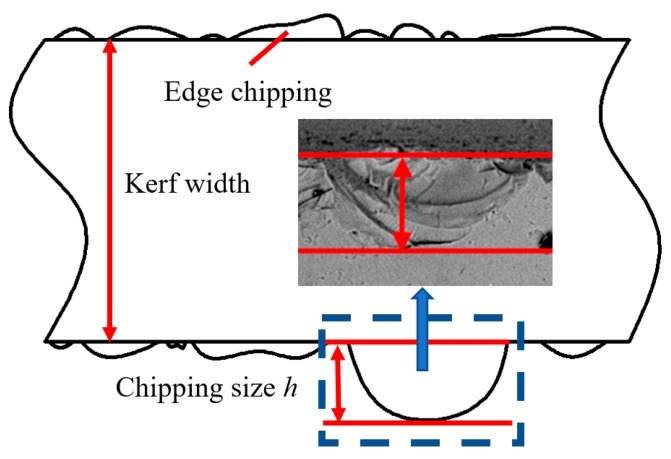
Measurement of edge chipping size.

**Figure 12 micromachines-10-00616-f012:**
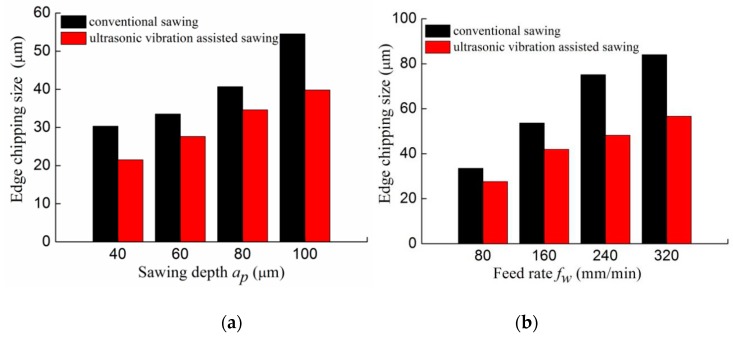
The edge chipping size versus sawing parameters: (**a**) The edge chipping size versus sawing depth (*v_w_* = 80 mm/min, *v_s_* = 10.5 m/s); (**b**) The edge chipping size versus feed rate (*a_p_* = 60 μm, *v_s_* = 10.5 m/s); (**c**) The edge chipping size versus line velocity (*a_p_* = 60 μm, *f_w_* = 80 mm/min).

**Table 1 micromachines-10-00616-t001:** Monocrystalline silicon properties.

Properties.	Density (g/cm^3^)	Crystal Orientation	Elastic Modulus (GPa)	Fracture Toughness (MPa·m^1/2^)
Value	2.33	100	130	0.8

**Table 2 micromachines-10-00616-t002:** The test parameters.

Parameters	Line Velocity *v_s_* (m/s)	Feed Rate *f_w_* (mm/min)	Depth *a_p_* (μm)	Ultrasonic Amplitude (μm)	Frequency *f* (kHz)
Value	10.5, 18.3, 26.2, 34.0.	80, 160, 240, 320	40, 60, 80, 100	4	28

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
