# Peer review of "Investigation on the Edge Chipping in Ultrasonic Assisted Sawing of Monocrystalline Silicon"

_micromachines, 2019, doi:10.3390/mi10090616_

Round 1

Reviewer 1 Report

The authors significantly improved the submission from the first version. I am satisfied with the modifications made.

Author Response

Dear reviewer:

On behalf of my co-authors, we thank you very much for giving us an opportunity to revise our manuscript entitled “Investigation on the Edge Chipping in Ultrasonic Assisted Sawing of Monocrystalline Silicon”. (ID: micromachines-584403). we have revised the whole manuscript carefully and tried to avoid any grammar or syntax error. In addition, we have asked several colleagues who are skilled authors of English language papers to check the English. We would like to express our great appreciation to you and other reviewers for comments on our paper. Looking forward to hearing from you. Thank you and best regards.

Yours sincerely,

Mr. Xu Zhu

Reviewer 2 Report

The vector in Eq.1 is not shown in the coordination of Fig. 1, and the origin of coordination is not specific to the derivation of Eq.1. Which make it difficult to understand the meaning of Eq.1, and also hard to verify this equation. the meaning of v(sub w) is not defined in Eq.1? the meaning of d(sub p) after Eq.2 is not depicted in Fig.2, that makes it vague.  The characters in Fig.3 is unclear in blue color. The test results shown in Fig.7 seems very obvious and intuitive to this field, it lacks significant findings with those three factors from (a) to (c).  Is there any other evidence (course findings) besides the normal force and the tangential in Fig.7 to compare the effects of ultrasonic sawing with the conventional one? In the first paragraph of conclusion: It is not specifically clear for the statement of "What has changed are mode changes in crack propagation." At the end of the first point in conclusion: " As one of ways that embody the impact of effective, the reduction of impact forces could directly decrease the size of transverse crack." is not specifically clear. And it would be better to be revised as "the ways..." and "the transverse..." Format of reference [37] should be corrected to be consistent.

Author Response

Dear reviewer:

    On behalf of my co-authors, we thank you very much for giving us an opportunity to revise our manuscript entitled “Investigation on the Edge Chipping in Ultrasonic Assisted Sawing of Monocrystalline Silicon”. (ID: micromachines-584403). We have studied your comments carefully and have tried our best to revise our manuscript according to the comments. Simultaneously, we have revised the whole manuscript carefully and tried to avoid any grammar or syntax error. In addition, we have asked several colleagues who are skilled authors of English language papers to check the English. We would like to express our great appreciation to you and other reviewers for comments on our paper. Looking forward to hearing from you. Thank you and best regards.

Yours sincerely,

Mr. Xu Zhu

Response to Reviewer 2 Comments

1.The vector in Eq.1 is not shown in the coordination of Fig. 1, and the origin of coordination is not specific to the derivation of Eq.1. Which make it difficult to understand the meaning of Eq.1, and also hard to verify this equation. the meaning of v(sub w) is not defined in Eq.1?

Response: We have deleted the vector in Eq.1. And the coordinate used to derive the Eq.1 have been redrawn. Furthermore, the meaning of v(sub w) is the federate speed of workpiece. We have defined it in the revised manuscript.

2.The meaning of d(sub p) after Eq.2 is not depicted in Fig.2, that makes it vague.

Response: The meaning of d(sub p) is the penetration depth of diamond abrasive during the ultrasonic assisted sawing. We have redefined it in the revised manuscript.

3.The characters in Fig.3 is unclear in blue color.

Response: We have modified it to white colour.

4.The test results shown in Fig.7 seems very obvious and intuitive to this field, it lacks significant findings with those three factors from (a) to (c).  Is there any other evidence (course findings) besides the normal force and the tangential in Fig.7 to compare the effects of ultrasonic sawing with the conventional one?

Response: The variation trend of forces was used to explain reasons for the decrease of crack propagation and edge chipping size. Simultaneously, based on the original foundation, we have increased the variation trend between the specific energy and material removal rate. Because the specific energy is an index to evaluate the energy that removed the material per unit volume during processing. It can reflect the interaction mechanism between abrasive particles and materials in the process of sawing.

5.In the first paragraph of conclusion: It is not specifically clear for the statement of "What has changed are mode changes in crack propagation." At the end of the first point in conclusion: " As one of ways that embody the impact of effective, the reduction of impact forces could directly decrease the size of transverse crack." is not specifically clear. And it would be better to be revised as "the ways..." and "the transverse..."

Response: We have re-written the first point of conclusion according to the reviewer’s suggestion. The effects of impact force on crack propagation and edge chipping size are emphasized.

6.Format of reference [37] should be corrected to be consistent.

Response: We are very sorry for our negligence of incorrect format. And we have checked the full article and revised the problems.
